Original research

# Recent innovations in long-term care coverage and financing: a rapid scoping review

Marilyn Macdonald [1], Lori E Weeks [1], Erin Langman [1], Sheri Roach,[1] Morgan X MacNeil,[1] Julie Caruso,[1] Andrea C Tricco [2,3,4] Ba' Pham,[5] Sharon E Straus,[2,4] Sujata Mishra,[4,6] Wanrudee Isaranuwatchai,[7,8] Gordon V Cormack,[9] Maura R Grossman,[9,10,11] Alexa R Yakubovich,[2,12,13] Arezoo Mojbafan,[1] Melissa Ignaczak,[1] Caron Leid,[14] Jennifer Watt [5], Susan Stevens,[15] Tayaba Khan,[14] Janet A Curran [1], Elaine Moody,[1] Ricardo Rodrigues[16]

For numbered affiliations see end of article.

**Correspondence to**
Dr Marilyn Macdonald;
marilyn.macdonald@dal.ca

## ABSTRACT

**Objectives** To identify, chart and analyse the literature on recent initiatives to improve long-term care (LTC) coverage, financial protection and financial sustainability for persons aged 60 and older.

**Design** Rapid scoping review.

**Data sources** Four databases and four sources of grey literature were searched for reports published between 2017 and 2022. After using a supervised machine learning tool to rank titles and abstracts, two reviewers independently screened sources against inclusion criteria.

**Eligibility criteria** Studies published from 2017–2022 in any language that captured recent LTC initiatives for people aged 60 and older, involved evaluation and directly addressed financing were included.

**Data extraction and analysis** Data were extracted using a form designed to answer the review questions and analysed using descriptive qualitative content analysis, with data categorised according to a prespecified framework to capture the outcomes of interest.

**Results** Of 24 reports, 22 were published in peer-reviewed journals, and two were grey literature sources. Study designs included quasi-experimental study, policy analysis or comparison, qualitative description, comparative case study, cross-sectional study, systematic literature review, economic evaluation and survey. Studies addressed coverage based on the level of disability, income, rural/urban residence, employment and citizenship. Studies also addressed financial protection, including out-of-pocket (OOP) expenditures, copayments and risk of poverty related to costs of care. The reports addressed challenges to financial sustainability such as lack of service coordination and system integration, insufficient economic development and inadequate funding models.

**Conclusions** Initiatives where LTC insurance is mandatory and accompanied by commensurate funding are situated to facilitate ageing in place. Efforts to expand population coverage are common across the initiatives, with the potential for wider economic benefits. Initiatives that enable older people to access the services needed while avoiding OOP-induced poverty contribute to improved health and well-being. Preserving health in older people longer may alleviate downstream costs and contribute to financial sustainability.

## STRENGTHS AND LIMITATIONS OF THIS STUDY

⇒ This study uses the WHO definition of long-term care (LTC), which includes care across the continuum for older adults and does not exclude studies based on the type of care provided.

⇒ This study identifies gaps in the existing literature and suggests areas for improvement, such as the importance of capturing equity measures.

⇒ Due to the design of the review, long-established LTC programmes and new initiatives that are yet to be evaluated may not be captured, as well as proprietary reports and government and health authority internal reports.

⇒ Some of the included reports analysed textual data from policy data in the absence of implementation data.

## INTRODUCTION

The proportion of the global population aged 65 years and above is projected to grow from 10% in 2022 to 16% in 2050.[1] Due to rapidly ageing populations, a greater prevalence of chronic conditions including dementia, gains in life expectancy, changing family dynamics and living arrangements, and shortages of paid and unpaid caregivers, the demand for long-term care (LTC) services continues to expand.[2] Governments worldwide are challenged to create and adapt public programmes to ensure that older adults have access to high-quality LTC services. Accordingly, many governments undertook LTC system reforms.

In most countries, LTC services comprise a fraction of all healthcare expenditures.[3] The bulk of costs is covered by governments and social insurance with most people cared for in their own homes.[4] In general, systems of public LTC coverage can be clustered into two main types: (1) universal publicly funded system and (2) a means-tested, safety-net system where public funding is available for those who cannot afford to pay for LTC. Universal LTC is available in many countries, including Japan, Germany, Luxembourg, the Netherlands, South Korea, Denmark, Finland, Norway and Sweden.[5] Other countries, such as England and the USA, rely heavily on a means-tested, safety-net system.[5] The two main types of public LTC coverage can also be used in combination, as in the case of Canada, Austria, Australia, France, Greece, Ireland, Italy, Spain, Poland, New Zealand and Switzerland.[5] A systematic review noted that each country features its own distinct public LTC insurance (LTCI) implementation techniques and that public LTCI in all countries must continuously adapt to local conditions to function sustainably.[6]

Researchers have noted the substantial variations in LTC funding approaches between low-income and high-income regions, as well as the difficulty in finding relevant indicators and measures that can be adapted to different settings. Through a scoping review, Seyede and colleagues identified several indicators (eg, OOP expenditures, geographic coverage and proportion of older people receiving LTC) to measure progress towards Universal Health Coverage in the context of population ageing, but they noted that many of these indicators would not be feasible in low-income and middle-income countries.[7]

For the purposes of this review, LTC includes service provision in the community, such as home care and support, rehabilitative care, palliative care, residential care homes, assisted living facilities, nursing homes as well as any other institutional setting.[8] LTC coverage not only includes care provided in facilities, but it also includes programmes such as cash benefits, community services in kind, day programmes, residential facility care and home care.[8] According to WHO, 'financial protection is achieved when direct payments made to obtain health services do not expose people to financial hardship and do not threaten living standards'.[9] Financial protection via Universal Health Coverage is also a WHO Sustainable Development goal.[10] Financial sustainability 'refers to the extent to which a given set of fiscal policies for LTC does not shift too large a financial burden on future generations (ie, intergenerational fiscal equity) and ensures that "ends meet"'.[11]

The focus of this scoping review is LTC funding, specifically the knowledge gap regarding what is known about recent LTC initiatives that directly address financing. Addressing the complex needs of the world's rapidly ageing population will require a focus on improving aspects of LTC service coverage, financial protection and financial sustainability. The primary objective of this scoping review is to identify, chart and analyse the literature on recent public initiatives (ie, reforms, policies and programmes) that have been undertaken to improve LTC coverage, financial protection and financial sustainability.

### Review question
What recent initiatives have been implemented to improve service coverage, financial protection and financial sustainability in the LTC sector for persons 60 years and older?

## METHODS
This rapid scoping review was conducted in accordance with the JBI methodology for scoping reviews[12] and adheres to the reporting standards outlined in the Preferred Reporting Items for Systematic Reviews and Meta-Analyses (PRISMA) extension for Scoping Reviews.[13] The WHO guide on rapid reviews was consulted to help tailor the rapid review methods to decision-makers' needs.[14] The Strategy for Patient-Oriented Research Evidence Alliance processes for integrated knowledge translation were followed.[15–18] The WHO Centre for Health Development (WHO Kobe Centre) contracted this review. An agreement of work plan was provided to the contractor in lieu of a published protocol.

### Search strategy
After a limited search for seed articles in MEDLINE, two librarians (EL, JC) created an initial search strategy for MEDLINE that was peer-reviewed by a third librarian using Peer Review of Electronic Search Strategies.[19] A librarian translated the search strategy for CINAHL, EMBASE and EconLit. Sources of grey literature included ClinicalTrials.gov, WHO International Clinical Trial Registry Platform and ProQuest Dissertations and Theses. Canada's Drug and Health Technology Agency Grey Matters database was also used to identify additional sources of grey literature under the category of 'health economics'. All search strategies are available in online supplemental file 1 and are reported using PRISMA literature search extension.[20]

### Inclusion and exclusion criteria
Studies published in any language, from January 2017 to 25 August 2022, that captured recent LTC initiatives for people aged 60 and older, involved evaluation and directly addressed financing were included. Eligible study designs included randomised control trials, non-randomised controlled studies, before and after studies, interrupted time series, quasi-experimental studies, qualitative studies, mixed methods, economic evaluation, systematic reviews and policy analysis reports. Reasons for exclusion of literature sources included failure to address coverage, quality, financial protection and financial sustainability, no evaluation component and study populations under age 60.

## Study screening and data extraction

Bibliographic records were identified for screening using the Continuous Active Learning tool which uses supervised machine learning to rank titles and abstracts from most to least likely to be of interest.[21] [22] Results of the database searches were uploaded to Covidence (Veritas Health Innovation, Melbourne, Australia), and duplicates were removed. Three reviewers independently reviewed the first 1500 titles and abstracts. Potentially relevant full-text articles were then screened by two team members. All discrepancies were resolved by discussion or a third reviewer. The results of the search and the screening process are reported in full and presented in a PRISMA 2020 flow diagram.[23]

A data-extraction form was developed and pilot tested using a random sample of five included reports. Data were extracted on study characteristics, outcomes and PROGRESS-Plus[24] [25] characteristics, where possible (see online supplemental file). Reports written in languages other than English were reviewed and data extracted by team members fluent in those languages.

## Quality appraisal

Although quality appraisal is typically not conducted for a scoping review, it was conducted at the request of the commissioner.[14] The appraisal was conducted by one team member and independently verified by a second team member. JBI methodological quality assessment tools were selected based on the designs of included reports.[26–28] Review articles were appraised using the Health Evidence Quality Appraisal Tool.[29] Discrepancies were resolved through team discussion.

## Analysis

Using descriptive qualitative content analysis, data were categorised according to the key outcomes of interest, coverage, financial protection and financial sustainability in LTC.[12] Online supplemental table 1 outlines study characteristics, including initiatives and quality appraisal of included reports.

## Patient and public involvement

Two public partners, coauthors, attended regular team meetings over a 6-month period and provided input into the design of the study. They also reviewed and commented on report drafts, tools and manuscript drafts, as well as dissemination activities.

## RESULTS

The search retrieved 51766 electronic database and grey literature records. Following screening at the title/abstract level, 100 reports were identified for full-text review (figure 1). Of these, 24 reports met the inclusion criteria for this rapid scoping review.

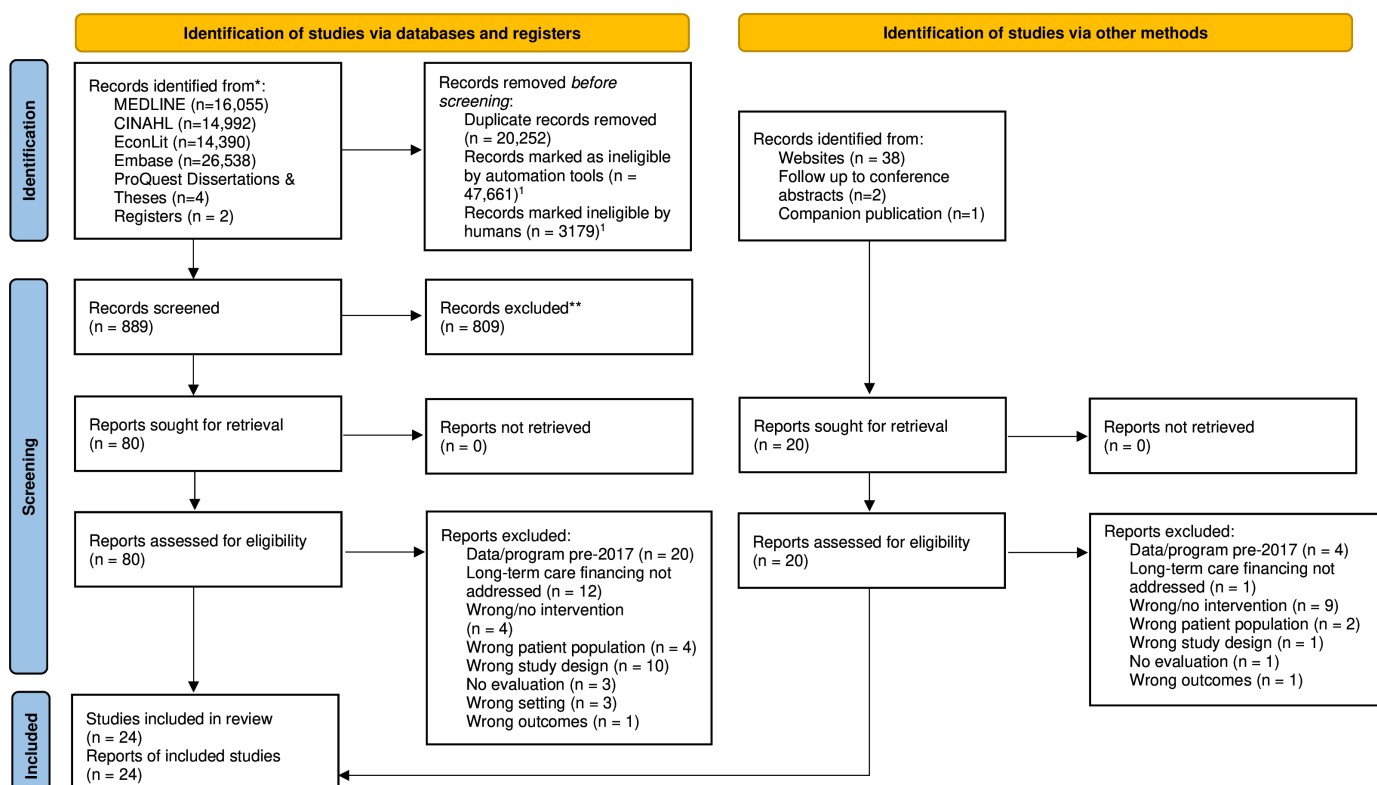

**Figure 1** PRISMA flow diagram of article selection. Three reviewers initially independently screened 500 records using CAL and a combined screening sample of 1970 for a total screening sample of 3470 records. With an estimate of duplicate records of 8.4% among the first 1500, we estimate humans screened a total of 3179 records using CAL. PRISMA, Preferred Reporting Items for Systematic Reviews and Meta-Analyses; CAL, Continuous Active Learning.

## Characteristics of included studies

Of the 24 reports, 22 were published in peer-reviewed journals, and 2 were grey literature sources.[30 31] Just over half of the reports addressed initiatives in China (n=13).[32–44] The remainder originated in the USA (n=3),[45–47] Singapore (n=2),[48 49] Taiwan (n=2)[50 51] and the Netherlands (n=1).[52] Three reports covered multiple countries.[30 31 53] All 24 reports were published between 2019 and 2022. The study designs included quasi-experimental study (n=7),[32 34 35 41 43 44 48] policy analysis or comparison (n=6),[31 38 42 49 51 53] qualitative research (n=3),[40 47 50] comparative case study (n=2),[30 45] cross-sectional study (n=2),[33 37] systematic literature review (n=2),[36 39] economic evaluation (n=1)[52] and survey (n=1)[46] (see online supplemental table 1).

Included studies analysed secondary data (n=10),[30–33 37 41 43 44 48 52] policy documents (n=6),[36 38 39 42 51 53] interviews (n=6)[33 40 45 49 50 53] and survey data (n=3).[35 46 47] One study had data from both policy documents and interviews.[53]

All included sources focused on people aged 60 and above, but not all included data collected from older adults. Six reports included data from LTC stakeholders.[40 45–47 50 53] For example, Beauregard and Miller conducted interviews with bureaucrats, consumer and provider advocacy groups, consultants and policy experts.[45] For these six studies, we have indicated both who the data concerned and who provided the data (see online supplemental table 1). Of the remaining 18 reports, 9[32–35 37 41 43 44 48] reported findings from a LTC population, and 9[30 31 36 38 39 42 49 51 52] focused on analysing policy, comparing LTC systems across countries or analysing government spending on LTC (see online supplemental table 1).

## Quality appraisal

21 out of 24 reports underwent critical appraisal. The three reports excluded from appraisal included two[30 31] summarising and interpreting existing studies, and the third[42] used a design that approximated a quality appraisal. Overall studies were appraised at moderate to high quality (see online supplemental table 1).

## Long-term care initiatives

The LTC initiatives are described in table 1.

## Equity

Although the term 'gender' was used consistently, only biological sex (female-male) was reported. Nine studies reported gender[32–34 37 40 41 43 44 48] (42.3% male). Place of residence was occasionally unclear. Education level as well as socioeconomic status and age were reported in multiple ways based on the study design and variables included. Online supplemental table 2 reports the PROGRESS-Plus characteristics reported by the study.

## LTC insurance coverage

The LTCI pilots in China represented LTC in the home setting and in institutions. The countries involved in the remaining initiatives have been delivering LTC in home care and have moved to either enhance initiatives or to reform existing initiatives. Table 2 outlines the five initiatives, associated studies and a summary of coverage by initiative.

### Initiative: LTC pilots, China

Six reports addressed coverage (table 2) where near-universal health insurance coverage is provided through three social health schemes: the Urban Employee Basic Medical Insurance, the Urban Resident Basic Medical Insurance and the New Rural Cooperative Medical Scheme for rural residents.[32 36 38–40 42] The pilots across 15 cities in China were funded through these existing schemes and largely covered the severely disabled.[32 36 40] Unmet care needs related to the activity of daily living (ADL) and instrumental ADL (IADL) were common in the absence of LTCI.[32] Seven of the pilot cities provided coverage for urban residents only, while eight expanded to varying degrees to provide coverage to rural areas as well as the non-employed.[39] Eligible participants are covered for both institutional and home care; however, in some cities, coverage for institutional care was more comprehensive than for home care.[39 40]

### Initiative: CareShield Life, Singapore

In 2019, Singapore implemented a mandatory social insurance scheme for citizens and permanent residents aged 30 and older born on or after 1 January 1980. Eligibility is determined by disability assessment (disability in performing at least three ADLs). CareShield Life provides lifetime coverage, a claim can be made at any age and premiums stop at age 67 years.[49]

### Initiative: LTC Plan 2.0, Taiwan

Taiwan's publicly funded LTC 2.0 Plan is directed to ageing in place, and institutional care is not included in the initiative.[51] The programme's successes included expanding coverage through the reduction of copayments from 30% to 16% and waiving these for low-income users. The plan also includes family caregiver support centres as well as dementia support centres for rapid diagnosis and access to services. Community care stations offering healthy living programmes focused on exercise and health self-management are part of the programme.[51]

### Initiative: Community First Choice, USA

Medicaid-supported LTC services and supports exclude many older people with a disability, and only 11% of older adults are Medicaid eligible (qualifiers include citizenship or permanent residency, income, household size, disability, family status and other factors).[47] Therefore, communities continuously fundraise to provide services to older people who do not qualify for Medicaid and cannot afford to purchase LTC services.[47] One recent initiative is the Community First Choice (CFC) Programme through which the federal government provides participating states with matching funding targeted at providing the services needed in the home setting to offset the need

**Table 1** LTC initiatives

| Initiative | Country | Author, year | Description |
|---|---|---|---|
| LTCI pilots | China | Chen *et al* 2020[33] Chen and Ning 2022[41] Dai *et al* 2022[39] Feng *et al* 2021[38] Han and Shen 2022[40] Lei *et al* 2022[32] Liu and Hu 2022[44] Peng *et al* 2022[42] Tang *et al* 2022[43] Wu *et al* 2020[37] Zhang and Yu 2019[35] Zhang *et al* 2020[34] Zhou and Dai 2021[36] | ▶ Nearly universal health insurance coverage yet did not include community LTC coverage before the pilots ▶ LTCI pilots instituted to learn and plan how best to deliver LTC ▶ Began in 2016 with 16 cities and expanded to 49 cities by 2020 ▶ Pilot cities explored multiple sources of financing and customised benefit packages based on their unique needs ▶ Aimed to promote formal (paid) care, while also encouraging informal care |
| CareShield Life | Singapore | Fong and Borowski 2022[49] | ▶ Legislated LTC social insurance ▶ Coverage for citizens and permanent residents born on or after 1 January 1980; entry age is 30, inclusive of people with severe disabilities and the economically disadvantaged ▶ Assessed for eligibility by the Ministry of Health (ie, disability in performing at least three ADLs) ▶ Lifetime coverage; a claim can be made at any age ▶ Benefits increase annually by 2% for the first 5 years; premiums stop at age 67 years; the later a claim is made, the higher the monthly payout |
| LTC 2.0 Reform | Taiwan | Chen and Fu 2020[51] Chiu *et al* 2019[50] | ▶ Expansion of LTC to target ageing in place ▶ Eligibility determined by assessing ADL, IADL, cognition, behavioural changes, rehabilitation, home situation and caregiver stress ▶ Eligible populations: people ≥65 years old with ADL limitations as well as frailty and people with dementia aged ≥50 years ▶ Changes included a reduction of copayments (meaning the premium amount paid for services), a simplified contract system for service providers, modified care worker service fees, government subsidies for rural areas, family caregiver support centres, dementia support centres and community care stations |
| Community First Choice | USA | Beauregard and Miller 2021[45] | ▶ The federal government programme allowing states to expand Medicaid home-based and community-based services through the Affordable Care Act ▶ 6% federal increase in matching payments to shift spending from institutional to home-based care ▶ Provides individuals considered to be at an institutional level of care with ADL and IADL assistance ▶ Agency-delivered or self-directed (clients take responsibility for hiring, supervising and dehiring workers) |
| LTC Reform | The Netherlands | Alders and Shut 2022[52] Hashiguchi and Llena-Nozal 2020[30] Kotschy and Bloom, 2022[31] | ▶ Reform initiated in response to high institutional care costs and focused on intensive home care as a substitute for institutional care ▶ Social care responsibility transferred to municipalities |

ADL, activities of daily living; IADL, instrumental activities of daily living; LTC, long-term care; LTCI, long-term care insurance.

for institutional care.[45] Many states are ideologically opposed to added government oversight and/or may be fiscally unable to meet matching funding to maintain the programme.[45]

**Initiative: LTC Reform, the Netherlands**

The Netherlands has a long history of comprehensive public coverage of LTC,[52] and a recent reform to address increasing costs involved shifting the financing of home

**Table 2** Coverage by country and initiative

| Initiative | Country | Author, year | Coverage | Coverage effect |
|---|---|---|---|---|
| LTCI pilots | China | Dai et al 2022[39] Feng et al 2021[38] Han and Shen 2022[40] Lei et al 2022[32] Peng et al 2022[42] Zhou and Dai 2021[36] | Eligible populations: seven pilot cities provided coverage to UEBM participants only. Eight cities expanded coverage to include both urban and non-employees of UEBMI and URBMI (n=2); urban employees enrolled in UEBMI and rural and urban residents aged 60 years and above enrolled in URBMI and NRCMS (n=1); all rural and urban participants in UEBM, UEBMI and URBMI (n=5) | Eligibility criteria are determined by the assessment of disability (assessment tools varied). People with severe disability were eligible in all pilot cities,[32 40] with some cities including coverage for moderate/mild disability (n=3)[32] and dementia (n=6)[40] |
| CareShield Life | Singapore | Fong and Borowski 2022[49] | Eligible population: citizens and permanent residents born on or after 1 January 1980 who are >30 years old | The overall assessment indicates that benefits are modest and that depth of coverage is sacrificed to breadth |
| LTC 2.0 | Taiwan | Chen and Fu 2020[51] | Eligible populations: people aged ≥65 years with ADL limitations as well as frailty and people with dementia aged ≥50 years | Eligibility criteria determined by assessing ADL, IADL, cognition, behavioural changes, rehabilitation, home situation and caregiver stress. LTC includes health prevention and supportive community services and excludes institutional care |
| Community First Choice | USA | Beauregard and Miller 2021[45] | Eligible populations: individuals who are Medicaid eligible. States that adopt the programme must provide services state-wide to everyone eligible | Beauregard and Miller found that states were concerned that this level of coverage would cause budgetary issues and that 'the added cost of CFC was not something that state officials were able or willing to shoulder' (p. 184)[45] |
| LTCI Reform | The Netherlands | Alders and Shut 2022[52] | Eligible populations: age 60 years and above. Social care transferred to municipalities, institutional care and federal responsibility and strict rules for eligibility | Municipalities with low economic solvency directed those eligible to institutional care |

ADLs, activities of daily living; CFC, Community First Choice; IADLs, instrumental activities of daily living; LTCI, long-term care insurance; NRCMS, New Rural Cooperative Medical Scheme; UEBMI, Urban Employee Basic Medical Insurance; URBMI, Urban Resident Basic Medical Insurance.

care away from the central government to municipalities, while maintaining the financing of institutional care. Municipalities where the fiscal status is sound are well situated to manage the shift, while those that are not find themselves in the position of directing people with home care needs to institutional care which is still financed by the central government. This was an unintended effect of the reform.

### Financial protection in LTC

11 reports addressed financial protection, 2 of which focused on OOP expenses.[30–33 40 41 43 48 49 51 52] One additional report contributed points relevant to financial protection.[38] Table 3 outlines financial protection in relation to the five initiatives.

### Initiative: LTCI pilots, China

Five reports directly[32 33 40 41 43] and one report indirectly[38] addressed financial protection in relation to the LTCI pilots in China. These reports addressed (OOP) health expenditures[32 41]; the relationship between LTCI and the presence of LTC services[33]; population groups without protection[40]; and LTCI impact on medical service utilisation.[43] OOP health expenditures were statistically

**Table 3** Financial protection outcomes by country and initiative

| Initiative | Country | Author, year | Financial protection outcomes |
|---|---|---|---|
| LTCI pilots | China | Chen and Ning 2022[41] Feng et al 2021[38] Han and Shen 2022[40] Lei et al 2022[32] Tang et al 2022[43] | ▶ LTCI significantly reduced inpatient OOP expenditure but not outpatient OOP expenditure. Individuals reporting fair health had higher OOP compared with those with poor health[41] ▶ Existing medical schemes are being used to finance the pilots; shifting of funds to LTCI could drive up OOP costs for URBMI and NRCMS enrolees[38] ▶ LTCI pilots do not financially protect rural residents, migrant workers and employees in new forms of employment. Across pilots, funding criteria vary with geography, occupation and age. LTC institutions offer greater financial protection than home care ▶ For each additional year of LTCI coverage, OOP medical expenses were reduced by 23.5%[40] ▶ Implementation of LTCI decreased outpatient expenses (22.82%) and inpatient expenses (19.8%) in pilot cities compared with non-pilot cities[43] |
| CareShield Life | Singapore | Fong and Borowski 2022[49] | Includes lifetime protection against catastrophic spending associated with severe disability. Although the scheme is described as providing good basic LTC care need protection, there is an expectation that over time the depth of the protection will develop based on need |
| LTC 2.0 | Taiwan | Chen and Fu 2020[51] | Changes to the LTC system included a reduction of copayments, reducing out-of-pocket spending for older adults |
| Community First Choice | USA | Beauregard and Miller 2021[45] Hashiguchi and Llena-Nozal 2020[30] | In California, the costs of home care for older people with median income, no net wealth and low care needs are not covered. It is also estimated that the public social protection system would not reduce the risk of income poverty for those earning a medium income with moderate care needs[30] |
| LTCI reform | The Netherlands | Alders and Shut 2022[52] Hashiguchi and Llena-Nozal 2020[30] Kotschy and Bloom 2022[31] | The use of asset testing means that those without net wealth would have lower out-of-pocket costs than the median income for older people. Those with net wealth would still be able to afford out-of-pocket costs from their incomes alone. Public support only covers help with ADLs—for help with IADLs or social activities, out-of-pocket costs can be up to 100%, depending on income and net wealth[30] |

ADLs, activities of daily living; IADLs, instrumental activities of daily living; LTCI, long-term care insurance; NRCMS, New Rural Cooperative Medical Scheme; OOP, out of pocket; URBMI, Urban Resident Basic Medical Insurance.

significantly reduced (23.5%) for inpatients but not for outpatients because OOP rates favour the institutional setting.[43] Existing medical schemes are being used to finance the pilots; shifting of funds to LTCI could drive up OOP costs for enrolees.[38]

Filial responsibility in China traditionally meant that family members fulfilled caregiving roles across the lifespan at rates bordering on 80%.[33] The initiation of LTCI is tied directly to the availability of LTC services for ADLs and IADLs in both rural and urban settings.[33] There was variability in who was considered eligible for LTCI. Groups such as migrant workers, individuals working in new forms of employment and rural residents were often excluded.[40] Over time, there has been a gradual expansion to include these groups.

Cities piloting LTCI have experienced a decrease in medical service utilisation due to a decrease in accessing both inpatient and outpatient services. Outpatient services use decreased by 22.8% and inpatient services by 19.8%.[43] At the same time, non-pilot cities saw these percentages rise. These reductions in medical system utilisation in the presence of LTCI hold promise for more optimal usage of the medical system, support for LTCI and thus a strategic shifting of the costs from medical to social care.

### Initiative: CareShield Life, Singapore

This initiative is an example of a universally publicly funded LTC scheme that extends to an entire population.[49] The scheme includes protection against catastrophic spending associated with severe disability and financial difficulty and nominal cash benefits with regular increases aimed at fostering consumer choice and flexibility, which can complement or substitute in-kind benefits and services. Although the scheme is described as providing good basic LTC care need protection, there is

an expectation that over time the depth of the protection will develop based on need.

### Initiative: LTC Plan 2.0, Taiwan

The reform of LTC undertaken in Taiwan was targeted towards assisting the population of older people to age in place.[51] The reform focused on the expansion of services that included but were not limited to dementia care, linking services such as adult daycare and respite services, family caregiver support centres, community health preventive care, links to discharge plans from hospitals and home-based medical care. The reduction in insurance copayments, availability of LTC support services and the subsidies provided to rural areas demonstrate the attention paid to financial protection.

### Initiative: Community First Choice, USA

In recognition of the number of people ineligible for LTC via Medicaid in the USA, the CFC Programme was conceptualised as an opportunity for individual states to consider protecting more of their older residents with social care needs from income poverty.[45] Hashiguchi and Llena-Nozal found that in California, even with social care systems in place, 'the out-of-pocket costs for older people with mean net wealth would be higher than their incomes' (p. 32) and that the costs of home care for those with median income, no net wealth and low care needs would not be covered, highlighting the risk of income poverty among older adults.[30]

### Initiative: LTC Reform, the Netherlands

The LTCI system in the Netherlands guarantees no older person experiences income poverty, despite OOP costs for low care needs,[30] because copayments are means tested to reduce the economic burden for the poor.[31] Municipalities that were unable to fund home-care recipients over the long term directed them to institutional care. Although institutional care would financially protect care recipients, those choosing to remain in their homes risk their financial protection. To counteract this, municipalities have been able to negotiate substantial reductions in maximum copayments since 2019.[52]

### Financial sustainability in LTC

Seven reports[34 36 40 42 45 49 52] directly address financial sustainability and three[38 39 50] peripherally. Table 4 portrays financial sustainability in relation to the five initiatives.

### Initiative: LTC pilots, China

Several studies[36 40 42] indicated challenges to the sustainability of China's LTCI pilots. The challenges reported were a lack of system integration,[36] reliance on transfers from the medical insurance fund[39 40] and a need for policy planning to address population ageing and align with economic development.[42] Feng et al advocated for the gradual increase in individual contributions in tandem with increased financial support from central and local governments to attain sustainability.[38] In addition, Zhang et al conducted a survey of care recipients

and their family caregivers in Shanghai to determine the impact of LTCI on informal care hours provided by family members.[34] Informal/family caregiver care was reduced by 0.47 hours. Across the 407 families, a weekly average decrease of 12.36 hours of informal care was reported. Satisfaction with formal care was 72%. These results speak favourably for the LTCI pilots to alleviate the burden on carers and the continued expansion of LTCI.

### Initiative: CareShield Life, Singapore

A policy analysis of CareShield Life which is a mandatory scheme spreads risk across the age cohorts, while including a prefunding component in which enrolees pay forward their future LTC needs, thus contributing to sustainability.[49] It is estimated that <0.1% of the population aged 30–40 years have severe disabilities; therefore, this insurance scheme is projected to not only meet LTC needs but also the needs of younger persons who may require services. These authors recognised the collectivist approach to this mandatory LTCI scheme, noting however that depth is sacrificed for breadth, and the need to conduct financial forecasting of the scheme to address sustainability.

### Initiative: LTC Plan 2.0, Taiwan

Chiu et al address coverage peripherally in a qualitative descriptive study of the implementation of LTC 2.0.[50] They found that the mechanisms for service integration needed to be fully implemented and also recommended building trust between service delivery agencies and government, clarifying roles and responsibilities of government care managers and agency care managers, the development of an integrated information system and the establishment of a budget tracking system.

### Initiative: LTC Reform, the Netherlands

LTC reform (in 2015–2019) was initiated in response to high institutional care costs and focused on intensive home care as a substitute for nursing home/institutional care. Social care responsibility in the home was transferred to municipalities. Alders and Schut conducted an economic evaluation of municipal-level data and estimated a random-effects model to explain variation across municipalities and to report the numbers of people admitted to LTC institutions annually.[52] The examination of municipality solvency was part of the evaluation. Municipalities with low solvency (20%) had 3.79 more admissions to public LTCI per 10 000 population over age 65 years, equivalent to 2.5% more admissions than average. The authors suggest that shifting can be addressed by risk adjusting their social support funding based on the proportion of older people by municipality and service needs. Such approaches optimise the use of funding sources and location of care and contribute to system sustainability.

### Initiative: Community First Choice, USA

The CFC programme offers a 6% federal increase in matching payments to shift spending from institutional

**Table 4** Financial sustainability by country and initiative

| Initiative | Country | Author, year | Financial sustainability findings |
|---|---|---|---|
| LTCI pilot | China | Dai *et al* 2022[39]<br>Feng *et al* 2021[38]<br>Han and Shen 2022[40]<br>Peng *et al* 2022[42]<br>Zhang *et al* 2020[34]<br>Zhou and Dai 2021[38] | ► Integrating LTCI into the existing medical insurance framework is favoured to reduce administrative costs, address basic LTC needs and gradually grow programmes as financing permits. 'The financial sustainability of the LTCI programmes has been questionable. A main reason for this is that the value allowed to be extracted from medical insurance foundations is limited' (p.196)[39]<br>► Diversification of funding sources such as cash benefits to support family caregivers and in-kind services may be necessary based on pilot funding in some cities. Such diversification is offered as a path towards sustainability[39]<br>► Glinskaya *et al* advocate for the gradual increase in individual contributions in tandem with increased financial support from central and local governments to attain sustainability[38]<br>► Funding sources affect sustainability, and most cities have more than one source. Primary reliance is on transfers from the medical insurance fund; other sources include individual contributions, organisational contributions, financial subsidies and social donations[40]<br>► The Policy Modelling Consistency Index was employed to measure the interaction coupling coordination degree between LTCI policy, population ageing and economic development to assess the sustainability of the LTC pilot system. Shanghai reached the excellent level, Nantong achieved good coordination and the remaining cities' coordination degree was basic. The coordination degree between LTCI policy, population ageing and economic development was reported as low or basic and suggests that economic development is not where it needs to be to sustain LTC financing. Sustained economic growth in tandem with population ageing and modifiable LTCI policies are essential to the financial sustainability of LTCI[42]<br>► A general lack of system integration was identified and believed to place financial sustainability in jeopardy[36]<br>► LTCI reduced informal/family caregiver care[34]<br>► Satisfaction with formal care was high[34]<br>► Continued expansion of LTCI is favoured[34] |
| CareShield Life | Singapore | Fong and Borowski 2022[49] | Policy analysis reported CareShield Life represents a collectivist (everyone contributes) approach to LTC social insurance, financial sustainability is guarded by rigorous claim criteria and eligibility assessments, inclusion of younger age groups contributes to affordability (coverage begins at age 30) and depth in service provision is sacrificed for breadth. Financial forecasting of the scheme is recommended to address sustainability |
| LTC 2.0 | Taiwan | Chiu *et al* 2019[50] | Examined difficulty with implementation of LTC 2.0 and challenges to sustainability |
| LTCI Reform | The Netherlands | Alders and Shut 2022[52] | LTC reform initiated to address high institutional care costs. Reform focused on intensive home care as a substitute for nursing homes/institutional care. Social care was transferred to municipalities. The unanticipated effect was a cost shifting by municipalities to the central government, which funds public LTCI (specifically institutional care). This economic evaluation study examined cost shifting with a focus on the solvency rate of municipalities. Results: municipalities with low solvency (20%) had 3.79 more admissions to public LTCI per 10 000 population over age 65 years, equivalent to 2.5% more admissions than average. Urban municipalities with low financial solvency, a higher population over age 80 years living alone or with limitations and with a lower income had a significantly higher number of admissions to the public LTCI system. Authors suggest shifting can be addressed by risk adjusting their social support funding based on the proportion of older people by municipality and on service needs. Such approaches optimise the use of funding sources and location of care and contribute to system sustainability |
| Community First Choice | USA | Beauregard and Miller 2021[45] | CFC was developed as an alternative to long-term institutional care. Participating states receive a 6% additional federal match to transition from Medicaid home-based and community-based service benefits. This comparative case study (two states with CFC and one without) examined factors that influenced adoption. Factors included leadership from the state Medicaid office, ability to meet the 6% matching, LTC advocacy and ideology related to federal government involvement.<br>A state government advocate stated, 'we have heard states really express concerns about the budgetary implications of this and whether it be sustainable even with the 6% increased FMAP (Federal Medical Assistance Percentage)' (p. 184) |

CFC, Community First Choice; LTCI, long-term care insurance.

to home-based care for individuals considered to be at an institutional level of care. Adoption and sustainability of the programme are contingent on federal home-based and community-based service policy that considers

implications at the state level. Higher federal match rates and reducing the state share of Medicaid spending would create the conditions for expansion and sustainability.[46] A case study of two states with the CFC programme

(Maryland and Texas) and one state without (Oklahoma)[45] reported that factors influencing the adoption of the CFC programme included state ideology, fiscal status, advocacy for LTC, Medicaid leadership and the strength of existing home-based and community-based services.

## DISCUSSION

13 of the 24 papers included in this review addressed LTCI initiatives in China. Coincidentally, China initiated 16 pilot studies of these initiatives in 2016, and evaluations of several of the pilots occurred in the 5-year window of our inclusion criteria for LTCI initiatives that were evaluated. The challenges and learnings from these pilots were common across the dataset and not deemed as unique to China. We found that, apart from the CFC programme, all initiatives either provided coverage to the entire population of the country or were working incrementally towards providing coverage to the entire population. Also common was the quest across initiatives to discern how to fund them. Older people with substantive care needs were the most likely to be eligible for services across all initiatives.

Although initiatives differed across the various countries, most entailed an increase in coverage. Strategies to increase coverage included expansion of eligibility criteria, reducing or waiving copayments based on income and simplifying processes for home support service providers. These strategies support ageing in place as do initiatives aimed at health promotion that assist older people in maintaining if not improving their physical and cognitive abilities.

The findings related to financial protection include strategies such as tackling OOP expenses and providing home support services to support ageing at home, mandating LTC, providing caregiver support and striking a balance between meeting LTC needs and how to fund them. These strategies saw a decrease in accessing both outpatient and inpatient medical services and access to LTC across the lifespan. A strategy that may help in the short and medium term is to develop an equitable LTC recipient dose sliding scale using ADLs, IADLs, cognition, behavioural changes, rehabilitation, home situation and caregiver stress. A sliding scale approach with individual contextualised person-focused assessments represents a strategy to push towards universalism, advance defamilisation and restrain marketisation[54] through optimal care to client matching. A sliding scale approach calls for LTC systems that are dynamic and that can increase or decrease supports and services based on changes in individual needs.

Financial sustainability was a goal across all initiatives. Evidence of this was found in the evaluation of the LTCI pilots that stressed the need for system integration, dedicated funding, controlling OOP costs and projecting LTC needs in relation to population ageing and economic development. Furthermore, Singapore mandated LTC and structured the scheme to enrol everyone at age 30 years, paying premiums until age 67 years and ensuring sustainability. Lastly, the Netherlands reform altered the funding plan to cost share with municipalities. In doing so, they discovered they needed to adjust the proportion of cost sharing based on the proportion of older people per municipality and service needs to address sustainability.

The findings of this review call forth the need to reflect on traditional approaches to LTC such as home care and institutional care. An alternate model could focus on enhanced services to support older adults to remain living in their homes and communities. A continuum that begins with remaining at home all the way to institutional care with more home-approximating options such as cohousing[55] may support optimal independence over a longer timeframe. This thinking is conceptually underpinned by the notion of providing the right support, in the right amount, at the right time to delay decline and to push down projections on the amount of care required, thereby contributing to financial sustainability. Such a continuum could contribute to positively addressing age-related inflation, which occurs when a group consumes more goods and services than it produces as they age.[56 57] The control of the need for services combined with an increasing workforce dampens inflation and improves the economic growth required to sustain health and social initiatives. Health promotion can slow the decline in health, thus delaying the need for supports and services.

## STRENGTHS AND LIMITATIONS

This team has breadth and depth in review methodology and the conduct of reviews as well as experts in LTC. The searches were extensive including four databases and multiple sources of grey literature, and the searches were peer reviewed. There were no language limits imposed on the searches to capture as many titles and abstracts as possible. This rapid scoping review was limited to identifying initiatives (programmes, policies or reforms) implemented and evaluated in the past 5 years in LTC for people aged 60 years and above. Countries need to include mechanisms within their initiatives to capture evaluation data. In the evidence assessed, LTC initiatives were focused on launching as opposed to evaluation, and we expect to see more evaluation including quality addressed in future research. Undoubtedly, there are initiatives that do not appear in this report because they did not meet all the inclusion criteria. In addition, there is commonly a lag between initiatives, their evaluation and subsequent publication. There is always the possibility that despite the number of database searches conducted, the keywords used in the searches may not have been used in a relevant publication. Many countries have long-established LTC programmes; however, unless a major modification occurred and met the inclusion criteria for this review, it would not have been captured. Some of the included reports analysed textual data from policy data in the absence of implementation data.

## Research gaps

We believe there are recent LTC initiatives in place that have not been evaluated or have been evaluated but not published and that such evidence needs to be published for all countries/subnations to benefit from. The number of economic evaluations of LTC initiatives was limited, and such evaluations should be encouraged to inform evolution in LTC. In addition to evaluation, implementation data from the initiatives are important to know what works and where improvements are needed to contribute to sustainability.

Although it may have been the intent of some researchers to capture equity measures, this was not necessarily made explicit in their reports. Absent from the data are race, religion and identification of any gender minority groups. Discussion on relating the characteristics to study results/findings was limited. Consistent use of a conceptual framework to compare policies such as that proposed by Noda *et al* is suggested to inform comparative evaluation studies or single initiative evaluation.[53]

## CONCLUSION

The initiatives related to coverage, financial protection and financial sustainability identified in this review came from high-income, upper-middle-income, lower-middle-income and low-income economies and hold hope for the growth of existing initiatives and their expansion within and beyond their respective countries.[13] The ones pertaining to lower-middle-income and low-income economies in particular fill a gap in a body of literature that has focused on high-income countries. Initiatives where LTCI is mandatory and accompanied by a commensurate funding scheme are situated to facilitate older people ageing in place. Efforts to expand population coverage are common across the initiatives, with the potential for wider economic benefits. In addition, initiatives that enable older people to access services while avoiding OOP-induced poverty contribute to improved health and well-being. Preserving health in older people longer may alleviate downstream costs and contribute to financial sustainability. The relevance and timeliness of LTC initiatives cannot be understated because by 2050, 80% of older people will live in low-income and middle-income countries.[58]

### Author affiliations

[1]School of Nursing, Dalhousie University, Halifax, Nova Scotia, Canada
[2]St. Michael's Hospital Li Ka Shing Knowledge Institute, Unity Health Toronto, Toronto, Ontario, Canada
[3]Dalla Lana School of Public Health, Epidemiology Division, University of Toronto, Toronto, Ontario, Canada
[4]Institute of Health Policy, Management, and Evaluation, University of Toronto, Toronto, Ontario, Canada
[5]University of Toronto, Toronto, Ontario, Canada
[6]Dalla Lana School of Public Health, University of Toronto, Toronto, Ontario, Canada
[7]Health Intervention and Technology Assessment Program, Mueang Nonthaburi, Nonthaburi, Thailand
[8]Unity Health Toronto, Toronto, Ontario, Canada
[9]David R. Cheriton School of Computer Science, University of Waterloo, Waterloo, Ontario, Canada
[10]School of Public Health Sciences, University of Waterloo, Waterloo, Ontario, Canada
[11]Osgoode Hall Law School, York University, Toronto, Ontario, Canada
[12]Department of Community Health and Epidemiology, Dalhousie University, Halifax, Nova Scotia, Canada
[13]Affiliate Scientist, Nova Scotia Health, Halifax, Nova Scotia, Canada
[14]SPOR Evidence Alliance, Toronto, Ontario, Canada
[15]Senior Director Continuing Care, Nova Scotia Health, Halifax, Nova Scotia, Canada
[16]ISEG Lisbon School of Economics and Management, Universidade de Lisboa, Lisboa, Portugal

**Acknowledgements** The authors are grateful to Dalhousie University Librarian Louise Gillis, who completed the peer review of the search strategy.

**Contributors** Design of the review: MM, LEW, ACT, EL, BP, GVC, MRG, SS, CL, TK, ARY. Data collection: MM, LEW, EL, AM, MI, BP, JC. Critical appraisal: MM, LEW, BP. Data analysis and interpretation: MM. Drafting the article: MM, EL, JC, SR, MXM. Critical revisions of the article: MM, LEW, EL, JC, ACT, BP, SS, SM, WI, GVC, MRG, ARY, AM, MI, CL, JW, SS, TK, DL, JC, EM, RR, SR, MXM. All authors approved the final version to be published. MM, guarantor.

**Funding** This research was supported by the WHO Centre for Health Development (WHO Kobe Centre: K22005).

**Competing interests** LEW is the co-owner of an assisted living home.

**Patient and public involvement** Patients and/or the public were involved in the design, or conduct, or reporting, or dissemination plans of this research. Refer to the Methods section for further details.

**Patient consent for publication** Not applicable.

**Ethics approval** Not applicable.

**Provenance and peer review** Not commissioned; externally peer reviewed.

**Data availability statement** Data are available in a public, open access repository. Data included in this review are in the public domain and can be accessed via peer-reviewed journals or from the websites of organizations.

**Author note** We wish to acknowldge the contribution of Desmond Loong, Health Economist to the review that led to this manuscript.

### ORCID iDs

Marilyn Macdonald http://orcid.org/0000-0002-0204-6278
Lori E Weeks http://orcid.org/0000-0001-5334-3320
Erin Langman http://orcid.org/0000-0002-3324-7152
Andrea C Tricco http://orcid.org/0000-0002-4114-8971
Jennifer Watt http://orcid.org/0000-0002-5296-6013
Janet A Curran http://orcid.org/0000-0001-9977-0467

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
