## [Reviewer comments · BMJ Open]

ARTICLE DETAILS

TITLE (PROVISIONAL)	Recent Innovations in Long-Term Care Coverage and Financing: A Rapid Scoping Review
AUTHORS	Macdonald, Marilyn; Weeks, Lori; Langman, Erin; Roach, Sheri; MacNeil, Morgan; Caruso, Julie; Tricco, Andrea; Pham, Ba; Straus, Sharon; Mishra, Sujata; Isaranuwachai, Wanrudee; Cormack, Gordon; Grossman, Maura; Yakubovich, Alexa R; Mojbafan, Arezoo; Ignaczak, Melissa; Leid, Caron; Watt, Jennifer; Stevens, Susan; Khan, Tayaba; Curran, Janet; Moody, Elaine; Rodrigues, Ricardo

VERSION 1 – REVIEW

REVIEWER	Lee-Ngow, Zemirah University of California San Diego, Library
REVIEW RETURNED	30-Aug-2023

GENERAL COMMENTS	I would like to thank the editor for giving me the opportunity to review this article. This article summarized initiatives from 2017-2022 to improve long-term service coverage, financial protection, and financial sustainability for persons 60 years or older. Comments: 1. This article followed internationally recognized reporting and assessment guidelines, such as the JBI methodology for scoping reviews and PRISMA. Additionally, the authors used a well-known evidence-based knowledge synthesis and translation method (SPOR-EA) to guide their review. These should be recommended and promoted to the field to improve the transparency of reproducibility of reviews. A published protocol was not registered, as it followed a previously agreed upon work plan contracted by the WHO Centre for Health Development (WHO Kobe Centre).2. I found this article to be well written with all key components to be neatly organized in each respective section of the manuscript. Of particular note, the searches conducted and documented by the Librarians (and peer-reviewed in PRESS and PRISMA-S) were fairly comprehensive and cleanly captured, following a logical progression, and conceiving as many variations as possible. I did note that the authors incorrectly switched the search strategy links between Medline and CINAHL respectively (See Supplemental Material Table 1, Search Strategies, and Data Extraction Tool.docx).a. https://www.cabidigitallibrary.org/doi/10.1079/searchRxiv.2023.00213 (CINAHL)b. https://www.cabidigitallibrary.org/doi/10.1079/searchRxiv.2023.00214 (Medline)
---

	3. The persons/population of interest should clearly be identified in the Abstract's Objectives (lines 31-33). See Review Question (113). 4. Inclusion criteria (line 136) is reported but not exclusion criteria. Why? Your PRISMA flow indicates reports excluded for reasons not mentioned in the inclusion criteria such as wrong setting and wrong outcomes. You also mention that studies did not meet the inclusion criteria but don't explain why (line 435) Defining your exclusion criteria would make this section stronger. 5. I did appreciate the considerations towards equity on race, religion, and identification of any gender minority gaps in the tone of this article and hope equity becomes more of a standard consideration in future scholarship.
--	--

REVIEWER	Ikegami, Naoki School of Medicine, Keio University
REVIEW RETURNED	31-Aug-2023

GENERAL COMMENTS	First, long-term can no longer be dichotomized into institutional care and community care. This is because the distinction between the two has become blurred with the expansion of special housing for those needing care. Second, the term "copayment" is used. But a distinction should be made between a proportion of the cost of services, and the cost of "hotel services" (bed and board). Third, extent to which public assistance is readily available for those who are not able to pay the copayment or the cost of hotel services. How the financial needs of the spouse are met if the person were to be hospitalized. Lastly, Japan, which has one of the largest LTC programs in the world is not included.
---

REVIEWER	Froehlich Chow, Amanda University of Saskatchewan, Canadian Centre for Health and Safety in Agriculture
REVIEW RETURNED	13-Sep-2023

GENERAL COMMENTS	I was impressed with the attention to detail throughout the entire article. I feel the authors did a great job of setting the stage and discussing why this scoping review is needed to contribute to the literature. The methods section was detailed and clear, a reader could easily conduct the review using the authors methods as a guide. All key documents were included in the appendix, which is essential for a scoping review. I thoroughly enjoyed reading both the results and discussion, which did a nice job of exploring and unpacking the results. I have only one suggested revision, along with the limitations sections, I think the authors should include a strengths section- this is an important part of any manuscript.
--

VERSION 1 – AUTHOR RESPONSE

Reviewer 1	
1. I did note that the authors incorrectly switched the search strategy links between Medline and CINAHL respectively (See Supplemental Material Table 1, Search Strategies, and Data Extraction Tool.docx).	1. "Supplemental Material" file has been updated with the correct links.

a. https://www.cabidigitallibrary.org/doi/10.1079/searchRxiv.2023.00213 (CINAHL) b. https://www.cabidigitallibrary.org/doi/10.1079/searchRxiv.2023.00214 (Medline) 2. The persons/population of interest should clearly be identified in the Abstract's Objectives (lines 31-33). See Review Question (113). 3. Inclusion criteria (line 136) is reported but not exclusion criteria. Why? Your PRISMA flow indicates reports excluded for reasons not mentioned in the inclusion criteria such as wrong setting and wrong outcomes. You also mention that studies did not meet the inclusion criteria but don't explain why (line 435) Defining your exclusion criteria would make this section stronger.	2. Age of population of interest added to the abstract line 20. 3. Exclusion criteria added to both the abstract lines 28-30 and in the methods section the header of Inclusion Criteria was modified line 132 to add exclusion at lines 138-140
Reviewer 2 1. First, long-term can no longer be dichotomized into institutional care and community care. This is because the distinction between the two has become blurred with the expansion of special housing for those needing care. 2. Second, the term "copayment" is used. But a distinction should be made between a proportion of the cost of services, and the cost of "hotel services" (bed and board). 3. Third, extent to which public assistance is readily available for those who are not able to pay the copayment or the cost of hotel services. How the financial needs of the spouse are met if the person were to be hospitalized. 4. Lastly, Japan, which has one of the largest LTC programs in the world is not included.	1. Thank-you for this comment and we completely agree. On page 3 under strengths, we state that we used the WHO definition of long-term care, which includes care across the continuum for older adults. I realize that we use the term institutional care but consider it as part of the continuum of care. 2. Page 10 in the table we added that co-payment means for services.

	3. Publicly funded long-term care insurance programs vary in the terms of means testing; however, programs everywhere are working toward supporting the family as a unit. Not sure if you are asking us to make an edit or not. Therefore, no changes made. 4. You are correct and we draw your attention to page 24 lines 437-440 to explain why.
Reviewer 3 1. I have only one suggested revision, along with the limitation's section, i think the authors should include a strengths section- this is an important part of any manuscript.	1. On page 23 we inserted strengths into the heading with limitations and outlined some strengths in lines 428-431.

VERSION 2 – REVIEW

REVIEWER	Lee-Ngow, Zemirah University of California San Diego, Library
REVIEW RETURNED	03-Jan-2024

GENERAL COMMENTS	Thank you for making the requested changes to your very informative and timely research.
--

REVIEWER	Ikegami, Naoki School of Medicine, Keio University
REVIEW RETURNED	08-Jan-2024

GENERAL COMMENTS	The authors have responded adequately to the comments raised by the reviewers. One aspect which I failed to notice in my first review is that of the 22 papers studied, 13 were from China. This may have had impact on the conclusions. This manuscript would be improved if a brief explanation were to be added concerning this issue.
---

VERSION 2 – AUTHOR RESPONSE

Comments	Responses
Reviewer 1 1. Nothing further	
Reviewer 2 1. The authors have responded adequately to the comments raised by the reviewers. One aspect which I failed to notice in my first review is that of the 22 papers studied, 13 were from China. This may have had impact on the conclusions. This manuscript would be improved if a brief explanation were to be added concerning this issue.	Thank-you for this observation. To address this comment, we will open the discussion with the following: Thirteen of the 24 papers included in this review addressed LTCI initiatives in China. Coincidentally China initiated 16 pilot studies of these initiatives in 2016 and evaluations of several of the pilots occurred in the five-year window of our inclusion criteria for LTCI initiatives that were evaluated. The challenges and learnings from these pilots were common across the dataset and not deemed as unique to China.